"

# Brief communication: How extreme was the thunderstorm rain in Vienna on 17 August 2024? A temporal and spatial analysis.

Vinzent Klaus[1,*], Johannes Laimighofer[2,*], and Fabian Lehner[1, 3,*]

[1]GeoSphere Austria, Vienna, Austria
[2]Institute of Statistics, BOKU University, Vienna, Austria
[3]Institute of Meteorology and Climatology, BOKU University, Vienna, Austria
[*]These authors contributed equally to this work.

**Correspondence:** Vinzent Klaus (vinzent.klaus@geosphere.at), Johannes Laimighofer (johannes.laimighofer@boku.ac.at), and Fabian Lehner (fabian.lehner@geosphere.at)

**Abstract.** On 17 August 2024, a single thunderstorm cell over Vienna, Austria, produced 107 mm of rainfall within two hours at the "Hohe Warte" weather station, which has recorded hourly precipitation since 1941—one of the world's longest-running sub-daily precipitation time series. This amount, with an estimated return period of approximately 700 years, is nearly twice as large as the second-largest event on record in the greater Vienna area. Comparing rain gauge data with a radar-based gridded dataset, we found that multiple events were missed by the station network. 100-year return periods of two-hour precipitation were 63 to 77 mm for rain gauge data and 61 to 90 mm for the radar-based data, underscoring the extremity of the 2024 event. Although the extreme precipitation time series shows no significant statistical trend, conditioning the return period on average mid-tropospheric temperature reduces the return period from 700 to 300 years, suggesting that climate change increases the frequency of such events.

## 1 Introduction

On 17 August 2024, an extreme precipitation event occurred in the northwestern parts of Vienna, Austria. A total of 107 mm of rainfall was recorded within two full hours, including 94 mm in a single hour, at Vienna's Hohe Warte weather station in a loosely built-up residential area. The event triggered local flooding and resulted in one fatality (dpa, 2024). It also set new all-time records for the highest accumulation of 2-hour and 1-hour rainfall at Hohe Warte (GeoSphere Austria, 2024a). Such extreme precipitation events have become more intense due to climate change (Haslinger et al., 2025) and are projected to intensify further, driven by increased saturation water vapor pressure at higher air temperature (Seneviratne et al., 2021). Quantifying the return periods of such sub-daily events is critical for water management, e.g. for dimensioning wastewater drainage systems. One obstacle in quantifying these extremes is the limited length of hourly precipitation records. Systematic sub-daily precipitation observations mostly began in the 1950s (Lewis et al., 2019), with only a few earlier exceptions (e.g. Hobart, Tasmania), which constrains the robustness of extreme precipitation estimates. The start of hourly precipitation mea-

surements at Hohe Warte dates back to 1941, providing a quality-controlled time series that enables a more robust assessment of sub-daily precipitation extremes in Vienna.

The convective systems that are responsible for these sub-daily precipitation extremes are typically short-lived and extreme rainfall is spatially confined to only a few km$^2$. This raises the question of whether existing networks of sub-daily rain measurements are sufficient to capture such small-scale events. Between 2001 and 2018, only 17.3% of radar-detected hourly heavy precipitation events > 25 mm in Germany were captured by the rain gauge station network (Lengfeld et al., 2020). In Austria, around 280 weather stations with sub-daily measurements are operated by GeoSphere Austria (GeoSphere Austria, 2024b), corresponding to one station per 300 km$^2$. The station density in Vienna is significantly higher with seven stations located within the city limits, equating to one station per 60 km$^2$. Additionally, several stations situated close to the city borders further increase the likelihood of thunderstorms being directly recorded by rain gauges.

Nevertheless, additional observations from weather radars are becoming increasingly important in hydrology (Lengfeld et al., 2020). A key advantage of radars is their ability to provide full spatial coverage, enabling the estimation of return levels even at ungauged locations (Panziera et al., 2016). By adjusting radar-derived precipitation estimates with in-situ gauge data, errors in purely radar-based rainfall estimates can be mitigated (Goudenhoofdt and Delobbe, 2009; Rosin et al., 2024). GeoSphere Austria's operational nowcasting tool, "Integrated Nowcasting through Comprehensive Analysis" (INCA) (Haiden et al., 2011), follows this approach by combining a rainfall field derived from spatially interpolated surface station data with a radar-derived precipitation field. The combined field is optimized so it matches the rain gauge measurements at the station's locations, while the remote sensing data provide most of the spatial structure of the field.

The objective of this study is to quantify the exceptional nature of the 2-hour precipitation event in Vienna on 17 August 2024. To this end, we proceed in three steps: (i) analyzing the long-term sub-daily precipitation record at Hohe Warte, (ii) comparing the event with observations from other stations in Vienna, and (iii) examining its spatial characteristics using the INCA data set. Our goals are guided by the following research questions:

- How extreme was the event at Vienna Hohe Warte, considering the exceptional 84-year time series of hourly precipitation?

- What are the return periods of such an event at other stations in the Vienna basin, and are there records of comparable events in the neighboring time series?

- Can similar events be identified in the radar-based INCA data set for Vienna?

- What are the estimated 100-year return levels of 2-hourly precipitation for the INCA data set, and do they align with the station data?

## 2 Data and methods

The study area encompasses Vienna and its surrounding stations, covering approx. 1700 km$^2$. This region has average annual precipitation ranging from 550 mm to 800 mm, with values increasing toward the Vienna Woods in the west (Isotta et al.,

2014). The rain gauges used are operated by GeoSphere Austria and the hourly data is openly accessible through the web
portal "GeoSphere Data Hub" (GeoSphere Austria, 2024a) (station metadata in Table S1 in the supplement). Most of these
semi-automatic weather stations ("TAWES") are classified as SYNOP stations. For each of the selected eleven stations we
analyzed the complete available record - from the onset of hourly rainfall measurements through the end of 2023 - and derived
the daily maximum of 2-hour precipitation to avoid selecting multiple, auto-correlated time steps within the same event.

As gridded precipitation field we used the analysis from the standard 15-minute INCA version. The data are available at
a 1 x 1 km spatial resolution for the period 2004-2023. To ensure consistency with hourly station data, the 15-minute values
were aggregated to full hourly totals (e.g., 00-01 UTC, 01-02 UTC). Furthermore, total precipitable water (PWAT; column-
integrated amount of water vapor from the surface to the top of the atmosphere) and average temperature between the pressure
levels 500 hPa and 700 hPa ("cloud temperature" as in Formayer and Fritz (2016)) were obtained from the ERA5 data set
(Hersbach et al., 2020).

Extreme value analysis was conducted by a peaks over threshold (POT) approach. Each time series (rain gauges and INCA)
was limited to the months May to September, and the $k * nyear$ largest maximum daily 2-hour precipitation events were
selected. We set $k = 3$, and $nyear$ denotes the number of available years of the respective time series. Missing station values
were permitted, as for ten stations the number of missing hourly values was below 5 %. For the estimation of return periods, a
regional frequency approach (RFA) was chosen (Hosking and Wallis, 1997), as preliminary results showed that this approach
yields the most robust results compared to single-site estimates by L-moments or maximum likelihood. In the RFA framework,
L-moments of the initial eleven stations were computed and weighted by the length of the station record for estimation of
the (regional) parameters of the Generalized Pareto distribution (GPD). Three stations with large deviations from the regional
L-moment estimates were discarded from the analysis to ensure a homogeneous study area. The 100-year return period for the
INCA data set was computed using the index quantile function and weighting it by the mean of the POT-series for each grid
cell. Confidence bounds (2.5 %, 25 %, 75 %, 97.5 %) for the eight remaining stations and each INCA grid cell were estimated
by a Monte-Carlo simulation. Finally, a distributional Bayesian regression was set up to account for a possible increase in
precipitation extremes conditional on cloud temperature. All statistical analyses were performed in R (R Core Team, 2024)
using the packages lmomco (Asquith, 2024), lmomRFA (Hosking, 2024), and bamlss (Umlauf et al., 2018).

For the event description, High-Resolution Visible (HRV) satellite images from MSG satellites (Schmetz et al., 2002) and
weather radar data (reflectivity and Doppler radial velocity) from the Austrian radar network operated by Austro Control
(Kaltenböck, 2012) were used.

## 3 Results and discussion

### 3.1 Synoptical situation and event description

On 17 August 2024, Central Europe was in a moderate southwesterly flow regime ($\approx$25 kn) between a 500 hPa trough extending
over Western Europe and a ridge over Eastern Europe (Figure S1a in supplement). At the surface, the pressure gradients were
weak and wind speeds in Vienna were around 5 kn from the Northeast, but changing to a southwesterly wind above 900 hPa.

Prior to the event, a 2 m dewpoint of 19 °C and a maximum 2 m air temperature of nearly 33 °C indicated a warm and humid boundary layer. The 12 UTC radiosonde measurement showed a surface-based Convective Available Potential Energy (CAPE) of ≈1500 J/kg and entrainment CAPE of ≈1100 J/kg (Figure S1b in Supplement). Moisture availability was exceptionally high throughout the troposphere, as evidenced by ERA5-derived PWAT that peaked at 45 mm during the event. The daily average on 17 August was 42.4 mm, which ranks as the third-highest daily mean value for August in the period 1941–2024. The 1991–2020 average for mid-August is 27 mm. From a climate change perspective, PWAT has been shown to increase by approx. 6 % per 1°C rise of 2 m air temperature in Europe (Wan et al., 2024). Furthermore, the 2-hourly average temperature between 500 hPa and 700 hPa was 273 K, which was the maximum value of all extreme precipitation events from the extreme value analysis, and only eight events recorded a temperature higher than 272 K. While the unstably stratified, warm, and humid troposphere provided favorable conditions for heavy thunderstorms, the vertical wind shear was rather weak (0-6 km shear of 21 kn), which hindered storm organization and updraft longevity.

Around 11:00 UTC (13:00 local time), HRV satellite images showed isolated thunderstorms forming over the Lower Austrian Prealps ≈ 60 km southwest of the city. By 12:00 UTC, towering cumulus clouds covered large parts of the Vienna woods, a hill range extending from the Prealps to the western city districts. At 13:00 UTC, multiple thunderstorm cells developed in this area. Shortly afterward, an outflow boundary originating from decaying thunderstorms approx. 100 km northwest of Vienna reached the city, establishing a convergence line between westerly and easterly winds along the foothills of the Vienna woods. The strong convergence contributed to the rapid development of a new thunderstorm cell in the northwest part of Vienna, first appearing on weather radar images at 13:45 UTC. TAWES measurements and Doppler radar data indicate that the convergence line remained nearly stationary for the next 20 to 30 minutes, during which the cell intensified significantly. The strongest rainfall at Hohe Warte was recorded from 14:20 to 14:40 UTC, with 10-minute rain totals of 24.1 mm and 26.7 mm, respectively. At the same time, a microburst produced maximum wind gusts of 40 kn at Innere Stadt, a weather station in the city center 5 km to the southeast of Hohe Warte. By 15:00 UTC, the storm cell began to weaken, but light stratiform precipitation from the dissipating storm cluster continued for another 3 hours. The highest 2-hour precipitation sum was 110 mm from 13:50 to 15:50 UTC, or 107 mm for the two full hours from 14:00 to 16:00 UTC.

## 3.2 Station-based analysis

Prior to 2024, only two events in 2014 (62 mm) and 2021 (58 mm) exceeded 50 mm within two hours at the station Hohe Warte (see Fig. 1a for daily maxima of 2-hour precipitation from 1941 to 2024). The block bootstrapped Mann-Kendall trend test by Önöz and Bayazit (2012) revealed no significant trend in the time series. In their respective observation periods, all eleven stations in the study area recorded only six events with 2-hour precipitation sums greater than 50 mm. Considering the maximum events of all eleven stations, these ranged from 38 mm (Stammersdorf, sub-daily measurements starting in 2008) to 62 mm (Hohe Warte, excluding the record-breaking event). This highlights that the August 2024 event was not only extraordinary at the Hohe Warte station but also unprecedented across any station in the Vienna basin.

In a second step, we estimated the return period of the event, which is visualized in Fig. 1b. Since the results for computing these extremes are highly sensitive to individual events, we calculated the return period based on a regional frequency analysis,

which should give a more robust estimate for such extremes. The year 2024 was excluded for each time series, and thereby the record-breaking event itself. This yields a return period of almost 700 years (699 years) for the station Hohe Warte. The interquartile range for a 107 mm/2 h event spans from 507 years (25 %) to 1383 years (75 %), whereas the confidence bounds range between 184 years (2.5 %) and 3424 years (97.5 %). Including the year 2024 in the analysis would still result in a return period exceeding 500 years (528 years). Estimating the return level of 107 mm/2 h for each of the other stations in the Vienna basin yields mean estimates ranging from over 473 years to over 1000 years. In comparison, a 100-year event would correspond to 63 to 77 mm.

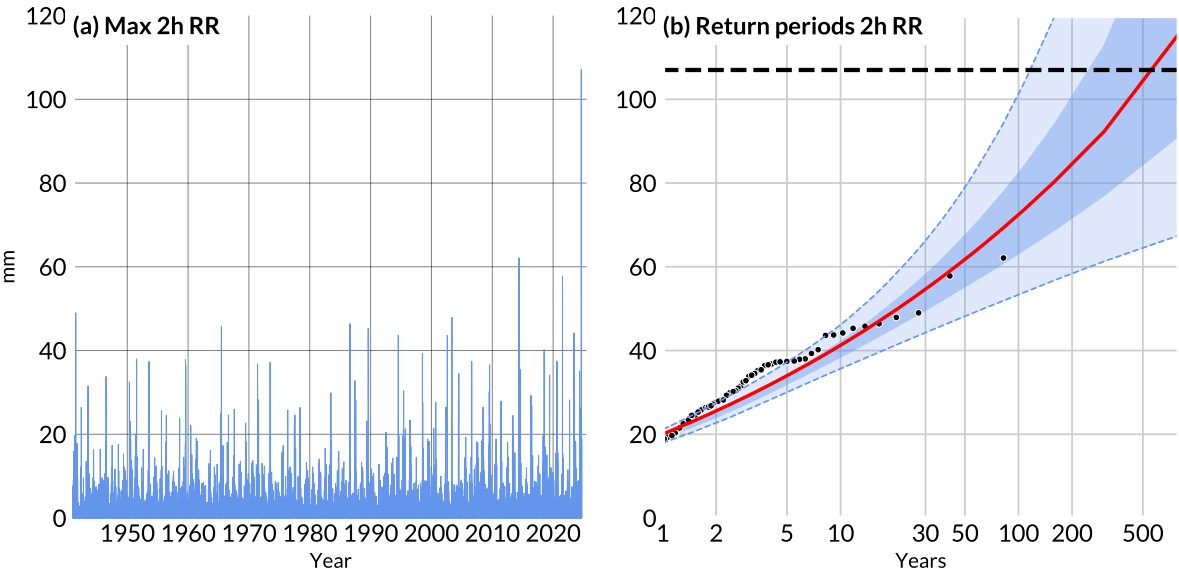

**Figure 1.** Extreme values statistics for station Hohe Warte. (a) Daily maxima of 2-hour precipitation totals. (b) Return period of 2-hour precipitation totals at Vienna Hohe Warte using observations from 1941-2024 based on the RFA approach. Light blue shading indicates the 95 % confidence interval, darker blue shading indicates the 25% and 75% percentiles and the red line the mean estimate. The horizontal dashed black line highlights the 107 mm from 17 August 2024.

Changes in extreme (convective) precipitation can also be linked to an increase in temperature due to the Clausius-Clapeyron scaling. Based on ERA5 reanalyses, mean summer cloud temperature (700 hPa to 500 hPa) has been increasing for Vienna, particularly since the 1980s. Estimating the return period for extreme precipitation conditional on cloud temperature can give additional insights into the future likelihood due to progressing climate change. This conditional approach yields a return period of approx. 300 years, which is substantially lower than the approx. 700 years for the stationary model, suggesting an impact of climate change on such events.

### 3.3 Spatiotemporal analysis of INCA

The station-based analysis indicated that 2-hour precipitation events larger than 50 mm are extremely rare in the Vienna basin. One possible explanation could be the low chances of a thunderstorm precipitation core being recorded by gauging stations, despite the high station density in Vienna. Potentially unrecorded events can be investigated using the INCA data set, which we will use in a threefold approach.

First, we analyze the spatial patterns of the five events with the largest 2-hour precipitation sums observed at gauging stations (Fig. 2). For the August 2024 event, it is evident that the station Hohe Warte was in the center of the precipitation core (Fig. 2a). Based on INCA, an area of $17 \, \text{km}^2$ was hit by more than 75 mm of rainfall, and only $4 \, \text{km}^2$ by more than 100 mm. For the events in 2014 (Fig. 2b), 2010 (Fig. 2d), and 2008 (Fig. 2e) at least one weather station was close to the precipitation hotspot. For the 17 July 2021 event (Fig. 2c), however, the precipitation center was not recorded by any station, and the INCA maximum of 98 mm exceeds the station maximum by 40 mm. We note that the 2-hour precipitation totals in this case originated from multiple distinct storm cells.

This leads us to our second analysis, which utilized INCA to investigate extreme events of 2-hour precipitation at the grid cell level. We found that no event in the period 2004-2023 surpassed a precipitation maximum of 100 mm, but six events exceeded 75 mm within two hours. The maximum area with precipitation above 75 mm was $16 \, \text{km}^2$ on 17 July 2021 (Fig. 2c). The strongest event not featured in the station-based analysis above was on 3 June 2020 with a maximum 2-hour precipitation of 87 mm south of Vienna and an area of $7 \, \text{km}^2$ exceeding 75 mm/2 h. For the period 2004-2023, we identified a total of 55 events > 50 mm/2 h. Compared to only six cases of > 50 mm/2 h recorded at weather stations, we conclude that the number of extreme events sampled by the stations is likely underrepresented. Further, only 15 events > 50 mm/2 h were observed by INCA in the years from 2004 to 2013, but 40 events from 2014-2023, indicating either a trend in extreme precipitation events or a possible inhomogeneity in the data set.

Lastly, we compare an extreme value analysis of the INCA data set to station data. A POT series of 2-hour precipitation sums is built for the INCA data set between 2004 and 2023. Due to the limited INCA time period we did not analyze the return period of the 107 mm/2h event itself, but evaluated 100-year return periods. The regionalized parameters of the GPD were used as the index rainfall distribution for each INCA grid cell. Note that not all grid cells may be well represented by the RFA approach. To assess this, the deviations of the L-moments from the regionalized L-moments were computed, and only about 11% of the grid cells exceeded the critical threshold of the discordancy measure (Hosking and Wallis, 1997). Fig. 3 shows the mean estimate and the confidence levels of a 100-year return level for INCA and the stations. Generally, the return periods of the station data align with those of the INCA data set. For the mean estimate, the INCA-derived 100-year return periods of 2-hour precipitation range from 60 to 91 mm within the domain, compared to 63 to 77 mm derived from station time series. The upper confidence bounds (97.5 %) of a 100-year event would already include an event like the one in August 2024 in the northern parts of Vienna.

Interestingly, the spatial pattern of 100-year return levels in the Vienna area shows no west–east gradient and thus does not reflect the climatological annual precipitation distribution - extreme precipitation events do not seem to be favored along

the hilly Vienna woods. Some recent studies have investigated the influence of urban areas on precipitation and found local enhancements in or downwind of big urban areas (McLeod et al., 2017; Torelló-Sentelles et al., 2024). Proposed mechanisms of this enhancement are changes in the local atmospheric circulation similar to a land-sea breeze and modifications of cloud and nuclei condensation through pollutants (Amorim and Villarini, 2024). However, no such effect has been documented for the Vienna area yet, and our results also do not allow conclusions in this regard.

Our spatiotemporal analysis highlights two key points. First, even in a relatively densely measured region like Vienna, INCA detects considerably more extreme events than the in-situ station network due to its full spatial coverage. Second, the upper confidence bounds of a 100-year event for INCA already encompass such an event, which could be attributed to the short record length of INCA. Nevertheless, our analysis demonstrated that this event was indeed extremely rare: neither the 84-year Hohe Warte record, the sub-daily precipitation measurements from Vienna's dense station network available since the 1990s, nor the 20-year radar-based INCA data set indicate that a comparable event was ever recorded in the Vienna basin.

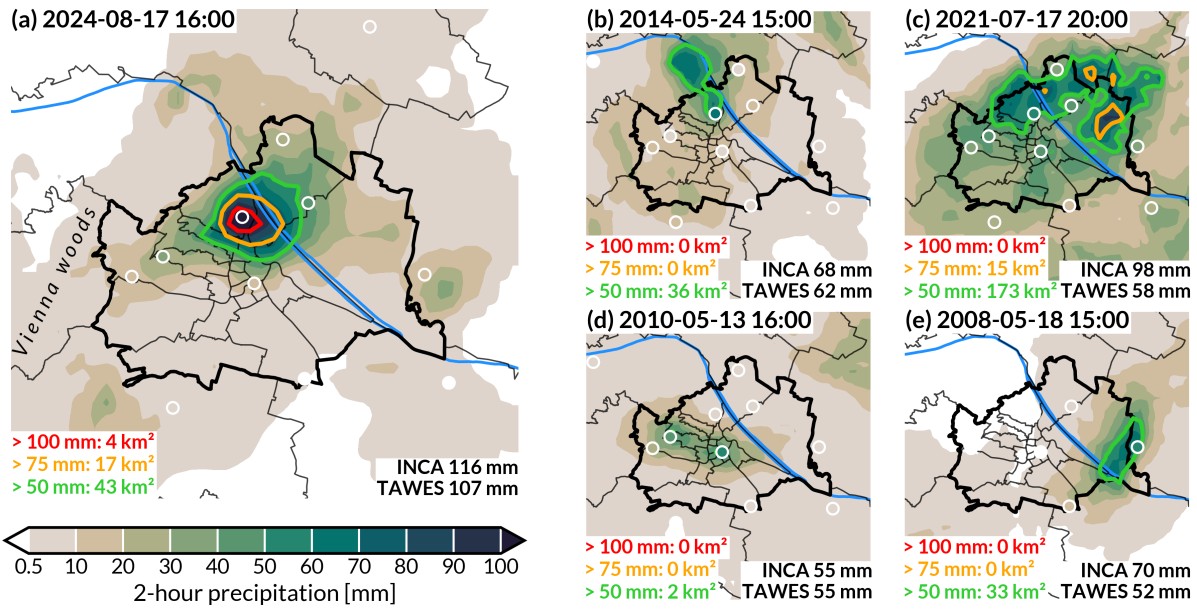

**Figure 2.** INCA precipitation totals (mm) for the five highest 2-hour rainfall events recorded at TAWES weather stations in Vienna and its surroundings since 2004. White-contoured dots indicate TAWES locations, with the dot fill color corresponding to the station rainfall measurement. Green, orange, and red contours indicate areas where 2-hour INCA rainfall exceeds 50 mm, 75 mm, and 100 mm, respectively. In the right bottom corner of each subplot, maximum values recorded at TAWES stations are contrasted with maximum INCA values.

## 4 Conclusions

We analyzed an extreme precipitation event in Austria that brought 107 mm within 2 hours when a thunderstorm's precipitation core hit Vienna's Hohe Warte weather station on 17 August 2024. The exceptionally long hourly time series from this station

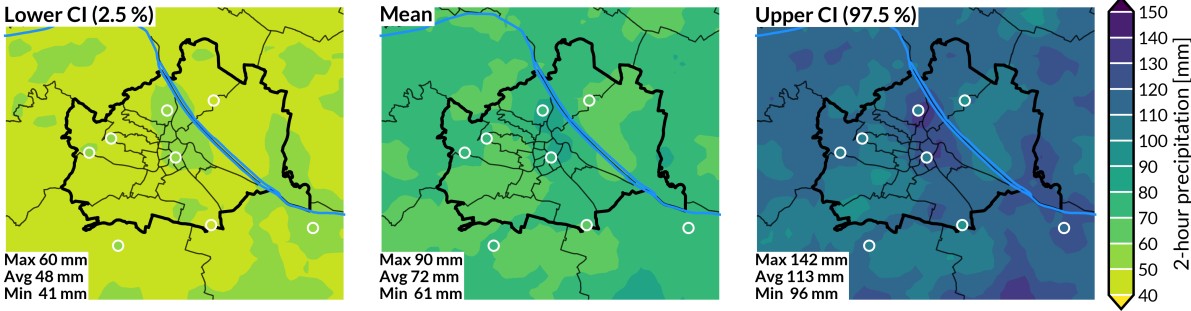

**Figure 3.** Estimation for a 2-hour precipitation event with a return period of 100 years for the INCA data set (2004-2023) and for each station represented by colored dots. A POT series was built for each pixel and station. The central panel shows the mean estimate, whereas the left (right) panel displays the lower (upper) confidence level (CI), based on a 95 % confidence interval.

dates back to 1941, and an analysis including neighboring stations shows that this event is unprecedented, with an estimated return period of approx. 700 years (25% percentile of 500 years and 75% percentile of 1400 years). No other event has ever exceeded 62 mm within two hours at any station in the study area. A comparison with the 20-year radar-based INCA data set confirms the found results from the stations. No such extreme event has occurred within the domain and the 100-year return values are within a similar range from both stations and INCA data set.

In the results we did not find significant trends of sub-daily precipitation extremes due to the high variability and mostly short records of sub-daily precipitation extremes, which in combination obscure a possible signal. In contrast, Haslinger et al. (2025) recently demonstrated that temporally smoothing the signal of extreme precipitation reveals a statistically significant increase of rainfall anomalies in Austria. Using this approach would yield similar results for our data set. As an alternative approach to assessing a potential climate change impact, we estimated the probability of the 2024 event conditional on cloud temperature, reducing the return period to 300 years. This is substantially lower than for the stationary model (700 years), suggesting that rising temperatures increase the intensity of extreme sub-daily precipitation in Vienna. This finding is in line with other studies such as Meyer et al. (2022) examining the trends in conditions favoring extreme precipitation events.

Our analysis demonstrates that including radar data yields additional insights into the return periods of extreme precipitation events. Historical events have been regularly missed by the existing rain gauge network, and their rainfall maxima were sometimes substantially underestimated. Consequently, the use of radar data is essential for obtaining more realistic estimates of extreme precipitation return periods, which are critical for public engineering and water management planning, such as designing sewage system capacity.

Limitations of this study arise from the length of the time series. Even though the weather station in question has 84 years of hourly measurements, the estimated return period of the event is much larger with a wide confidence interval. The same is true for the INCA data set used for the spatial analysis, which only spans 20 years. Additionally, INCA is primarily a nowcasting product. While it is expected to outperform purely radar-derived precipitation estimates in climatological analyses, it may suffer

from spatial or temporal inhomogeneity, e.g. changes in the INCA code base may have introduced temporal inhomogeneities
205  (Panziera et al., 2016).

*Code and data availability.*  Code for reproducing the data analysis can be found under https://github.com/katelbach/precipAnalysis. The weather station data and the aggregated INCA data used in this study can be downloaded at https://doi.org/10.5281/zenodo.14500708. Please note that the publicly available INCA data on the GeoSphere Data Hub is the 1-hour version and therefore differs from the 15-minute version used in this study.

210  *Author contributions.*  All authors contributed equally to the research design, data analysis and drafting the manuscript.

*Competing interests.*  The authors declare that they have no competing interests.

*Acknowledgements.*  We thank Jasmina Hadzimustafic from GeoSphere Austria for her support with the INCA data set and Marco Kopecky from GeoSphere for his information about the TAWES weather station network. This research has been supported by the Austrian Climate and Energy Fund under the program "ACRP13" (grant no. KR20AC0K17974).

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
