# Peer review of "Brief communication: How extreme was the thunderstorm rain in Vienna on 17 August 2024? A temporal and spatial analysis."

_Natural Hazards and Earth System Sciences, 2024_

## Author Comment (AC1)

**Response to Reviewer 1 (*Brief communication: How extreme was the thunderstorm rain in Vienna on 17 August 2024? A temporal and spatial analysis.* )**

The manuscript provides an analysis of the rainfall event on 17 August 2024 in Vienna, Austria. To assess the extremeness of the event two datasets have been used: Rain gauge data from weather stations with long records but limited spatial representativeness and the radar-based INCA dataset with spatial coverage but short time series. Both datasets revealed that the event on 17 August 2024 was extraordinary with a rainfall of 107 mm/2h and return periods in the range of several 100 years.

The study fits in the scope of the brief communication format of NHESS. It is well written, the goal and methods are clearly explained and the results are easy to follow and well illustrated. I recommend publishing the manuscript after addressing the following remarks:

We thank you for your insightful comments and we would like to address your discussion points and suggestions below.

P.3, L.54: What would happen in the unlikely case of an event around midnight? Is the independence of events still guaranteed?

As you mention, this case is extremely rare due to the pronounced diurnal cycle of thunderstorm activity. However, we still account for it by selecting only the higher 2-hour maximum if an event occurs in the two hours around midnight.

P.3, L.63: Isn't the INCA dataset also shorter than 25 years (2004-2023)? Why were annual maxima used instead of POT?

Thank you for your remark. We agree that this approach was inconsistent and updated the analysis. We now use a peak over threshold method for all time series (INCA and station-based analysis). For each time series, we selected the $k * nyear$ largest precipitation sums, where $k$ is 3 in our case, and $nyear$ is the number of available years of the specific time series. Additionally, the analysis is now based on a regional frequency analysis approach. For further details, please see our comments to Reviewer 2.

P.3, L.68: Please explain the abbreviation HRV, because it is used here for the first time.

We added "High-Resolution Visible" to explain the abbreviation.

P.3, L.72: I guess, "trough" is meant here instead of "through"?

Thank you, this typo is corrected.

P.4, L.94: Inner Stadt may not be clear to a read who isn't familiar with the city of Vienna. Maybe the authors could explain a bit more about the location.

We added that Innere Stadt is a weather station in the city center about 5 km to the southeast of Hohe Warte.

P.4, L.108: Just out of curiosity: How would the return period of the event change when it would be included in the time series?

For our updated estimates, this would yield a return period of about 520 years instead of 700 years if the year 2024 is left out.

40 P.5, Fig. 1b: A line indicating the 25% and 75% percentile would be helpful to show because it is mentioned in the text.

The 25 % and 75 % percentile were added to Fig. 1.

---

## Author Comment (AC2)

**Response to Reviewer 2 (Francesco Marra) (*Brief communication: How extreme was the thunderstorm rain in Vienna on 17 August 2024? A temporal and spatial analysis.* )**

45 This manuscript examines an extreme convective event that occurred in Vienna in 2024. It estimates the return period of the 2-hour rainfall in relation to the gauge location, to other gauges in the area and to a record of INCA model simulations. I find the study interesting and worth publication, although I think some weaknesses need to be addressed beforehand. Please see my comments below. I'll be happy to discuss with the authors about any misunderstanding from my side. Kind regards, Francesco Marra.

50 We thank you for your insightful comments and we would like to address your discussion points and suggestions below. Some of our responses are quite extensive, so we outsourced them to an external document.

Line 5: this kind of accuracy on an estimate of a very long return period is misleading. I suggest to write something like "likely exceeds 600 years".

55 We updated our estimation procedure, which now gives a return period of 699 years. In the text, we now refer to a return period of "approximately 700 years". We changed this throughout the text.

Line 65: please use "5%" or "five percent"
It is changed to 5%.

60

The temperatures during this summer event were particularly high, and so was the atmospheric moisture content. This likely contributed to the development of such an extreme event (as the authors mention in the introduction in line 13). It is true that the timeseries does not reveal a significant trend (and this is perfectly normal even in presence of trends, e.g., see https://doi.org/10.1029/2010WR009798), but it is natural to imagine that the ongoing rise in temperature likely contributed to
65 this extreme (and may contribute more in the future). I think this aspect cannot be neglected in the discussion and conclusions. Also, this is an important aspect to mention when assuming identical distribution in the extreme value analysis (we likely violate this assumption and this should be mentioned).
Thank you for this comment. First, our time series does not show any trend, so from a statistical point of view, the iid assumption is correct. Nevertheless, we agree that this may be an issue due to the "short" time series, or more specifically, the low signal-
70 to-noise ratio. In a simulation study (please find a detailed comment here: github.com/katelbach/precipAnalysis), we found that mean annual maximum precipitation would have to increase about 8% per decade to find a significant trend in 95 % of the cases. Assuming a monotonic trend - similar to our time series of 3.75% (per decade) - yields a significant trend only in 46% of the cases.

Recently, Haslinger et al. (2025) found an increase in extreme hourly precipitation for Austria. Their smoothing window
75 approach would yield similar results for our dataset, indicating a possible trend that is covered by the large variability in extreme precipitation and short-time records.

An additional possibility would be to link the data to (cloud) temperature data to detect a possible trend, including a covariable. Cloud temperature (average temperature in the 700 to 500 hPa layer as in Formayer and Fritz (2017)) in summer months shows a clear trend since the 1980s, but a possible positive bias before the 1970s (Fig. 1b as the 2m temperature before the 1970s features a different trend. This positive bias may be caused by missing or biased radiosonde data at the beginning of the time series, especially the 1940s, but a more detailed analysis is out of the scope of this manuscript.

Nevertheless, linking cloud temperature to extreme precipitation can give us an indication of the increasing likelihood of such an event due to rising temperatures. For the event in August 2024, such an estimate (a distributional Bayesian regression approach) would yield a return period of approximately 300 years, conditional on the estimated cloud temperature for the event.

All these issues were added in a shortened form to the manuscript for a broader discussion of the event and its underlying data.

[Figure]

(a) Mean annual cloud temperature in the summer months in Vienna from 1940 to 2024.

[Figure]

(b) Annual summer temperature at Vienna Hohe Warte from 1938 to 2024. Publicly available HISTALP dataset based on Auer et al. (2007).

Section 3.2: I think that here two aspects need some more attention

– The assumption of normality for MLE can in some occasions be a bit stretched. Why not using the same method to estimate uncertainty for both LM and MLE (i.e., the bootstrap)? This would grant consistency in the estimated confidence limits. Otherwise I don't think you should compare them.

We updated our estimate procedure by using a regional frequency approach (RFA). The RFA is estimated by L-moments weighted by the station record at each site. INCA estimates are based solely on their index value and are not included in the RFA. This approach now yields narrower confidence bounds for our estimates. We updated the text accordingly.

– given the importance of this section in the paper, I am a bit unsatisfied with only having estimates with these two parameter estimation methods (with uncertainties estimates in a rather inconsistent manner - see above). This is because 84 years can look like a long record but when it comes to extremes this is not really the case (you can just do some simple Montecarlo experiment from known distributions to see it)I don't want to bias the authors toward a specific method or distribution so I'll only give two classic examples. Many studies found that precipitation has extremes with GEV tail parameter close to 0.1. There are ways to include this information in the estimation (e.g., Martins & Stedinger https://doi.org/10.1029/1999wr900330). Another option is to use a regional framework such as the classic one based on LM by Hosking & Wallis (https://doi.org/10.1017/cbo9780511529443).

Please see our comment above.

– One of the take-home is that the event was really anomalous also considering that no trends could be seen in the data. An aspect that would be interesting to explore is: "assuming your GEV estimated before the event is true, what is the probability of observing this event in a 84 year time interval?" This can be done using Montecarlo sampling and can provide some useful insights.

Thank you for this hint. If we assume that our underlying distribution is correct, we can directly plug the probability estimate $(1 - P(x))$ in a binomial distribution and get a probability of 11.3 %, discovering this event at least once in 84 years. A Monte-Carlo simulation would converge against this value for large $n$. We added this aspect in the Discussion section.

The occurrence of such an extreme event over the city also raises questions about the possible impact of the urban area on extreme rainfall. Perhaps this aspect could be discussed - there is some literature on the topic.

Thank you for this comment. We added a paragraph discussing the possible influence of urban effects on this heavy precipitation event.

In line 169 you talk about possible temporal inhomogeneities in the INCA. I wonder if that is also a possibility in the 84 years gauge timeseries. I believe the instrument was changed over this time. Perhaps this aspect should be discussed.

The instrumentation at Vienna Hohe Warte has changed over the years. The measuring device was a floater when sub-daily measurements commenced in the 1940s. It was changed to a tipping bucket in the 1990s and further to a weighing gauge in

the 2010s. According to Haslinger et al. (2025) https://doi.org/10.1038/s41586-025-08647-2, the measurement bias is -5 % for floater devices, -3% for tipping buckets and -1% for weighing gauges. Considering these scales, the main statements in this paper would not change significantly.

125    I found a bit of a mismatch between the rarity of the studied event (over 500 years estimated return period) and the 20-year level used for the INCA estimates. I wonder if using different extreme value methods (e.g., see above two examples) it would be possible to look for longer return periods in the INCA. In general, we expect many ( 4) of the events in the gauge record to exceed the 20-year level, so this level is not that indicative for the 2024 event.

We agree and updated the plot to a 100-year event.

**References**

Auer, I., Böhm, R., Jurkovic, A., Lipa, W., Orlik, A., Potzmann, R., Schöner, W., Ungersböck, M., Matulla, C., Briffa, K., Jones, P., Efthymi-
adis, D., Brunetti, M., Nanni, T., Maugeri, M., Mercalli, L., Mestre, O., Moisselin, J., Begert, M., Müller-Westermeier, G., Kveton, V.,
Bochnicek, O., Stastny, P., Lapin, M., Szalai, S., Szentimrey, T., Cegnar, T., Dolinar, M., Gajic-Capka, M., Zaninovic, K., Majstorovic,
Z., and Nieplova, E.: HISTALP—historical instrumental climatological surface time series of the Greater Alpine Region, 27, 17–46,
https://doi.org/10.1002/joc.1377, publisher: Wiley, 2007.

Formayer, H. and Fritz, A.: Temperature dependency of hourly precipitation intensities–surface versus cloud layer temperature, International
Journal of Climatology, 37, 1–10, 2017.

Haslinger, K., Breinl, K., Pavlin, L., Pistotnik, G., Bertola, M., Olefs, M., Greilinger, M., Schöner, W., and Blöschl, G.: Increasing hourly
heavy rainfall in Austria reflected in flood changes, Nature, 639, 667–672, 2025.

---

## Author Response (AR2)

"

**2nd Author Response for Manuscript: *Brief communication: How extreme was the thunderstorm rain in Vienna on 17 August 2024? A temporal and spatial analysis.**

Vinzent Klaus[1,*], Johannes Laimighofer[2,*], and Fabian Lehner[1, 3,*]

[1]GeoSphere Austria, Vienna, Austria
[2]Institute of Statistics, BOKU University, Vienna, Austria
[3]Institute of Meteorology and Climatology, BOKU University, Vienna, Austria
[*]These authors contributed equally to this work.

**Correspondence:** Vinzent Klaus (vinzent.klaus@geosphere.at), Johannes Laimighofer (johannes.laimighofer@boku.ac.at), and Fabian Lehner (fabian.lehner@geosphere.at)

**Response to Editor comment (Oct 22, 2025)**

Dear Vinzent Klaus, Johannes Laimighofer, and Fabian Lehner!

Many thanks for the careful revisions made for this manuscript. Nevertheless, may I please ask you to work on one aspect to further improve the accessibility of the results. In the abstract, you mention that the 107mm/2hrs observed precipitation rate would be estimated as having a return period of about 700 years. For INCA, you give the 100-yr return periode as about 61-90mm/2hrs. Can these two pieces of information be presented in a more integrated way e.g., by giving the 100- or 700-yr RP for both respective datasets? Can this be reflected in the text more precisely?

Looking forward to this slight revision.

All the best

Gregor Leckebusch

Dear Gregor Leckebusch,

thank you for this helpful comment, which further improves the overall structure of the manuscript. We have included the 100-year return period for both INCA and the station data. However, we refrained from showing the 700-year return period for INCA, given the relatively short duration of the dataset (only 20 years). We hope this decision is in line with your expectations.

Sincerely,

The Author Team *(Vinzent Klaus, Johannes Laimighofer, and Fabian Lehner)* .